# Effect of Olfactory and Gustatory Dysfunction and Motor Symptoms on Body Weight in Patients with Parkinson’s Disease

**DOI:** 10.3390/brainsci10040218

**Published:** 2020-04-07

**Authors:** Carla Masala, Francesco Loy, Raffaella Piras, Anna Liscia, Laura Fadda, Alan Moat, Paolo Solla, Giovanni Defazio

**Affiliations:** 1Department of Biomedical Sciences, University of Cagliari, SP 8 Cittadella Universitaria, 09042 Monserrato, Italy; floy@unica.it (F.L.); raffaepiras@tiscali.it (R.P.); annamarialiscia@gmail.com (A.L.); 2Department of Neurology, Movement Disorders Center, Institute of Neurology, Azienda Ospedaliero Universitaria (A.O.U.), University of Cagliari, SS 554 km 4.500, 09042 Cagliari, Italy; fadda_laura@yahoo.it (L.F.); paosol29@yahoo.it (P.S.); giovanni.defazio@unica.it (G.D.); 3Medical Faculty, University of Cagliari, SS 554 km 4.500, 09042 Cagliari, Italy; moatalan@tiscali.it

**Keywords:** weight change, olfactory dysfunction, gustatory dysfunction, Parkinson’s disease

## Abstract

Background: Non-motor symptoms in Parkinson’s disease (PD) are often associated with a negative impact on the patients’ quality of life and on their weight regulation. The aim of this study was to assess the effect of olfactory and gustatory dysfunction, apathy, fatigue, depression, and motor symptoms on weight regulation in PD patients. Methods: We analyzed 112 participants, 63 PD patients (mean age ± SD: 69.2 ± 10.1), and 49 controls (mean age ± SD: 68 ± 9.6). For each participant we collected age, weight, height, BMI, olfactory and gustatory function, cognitive performance, apathy and fatigue. Results: Our data showed that 61.9% (*n* = 39) of PD patients had hyposmia, while 38.1% (*n* = 24) had anosmia. In PD patients, we observed a significant effect of Unified Parkinson’s Disease Rating Scale (UPDRS), apathy, odor threshold, sweet perception and fatigue on weight regulation. Instead, there was no significant effect for depression and levodopa equivalent daily dosage (LEDD). Conclusion: Our results suggest that PD non-motor symptoms such as olfactory/gustatory deficits and mood disorders may influence body weight.

## 1. Introduction

PD is a chronic neurodegenerative disorder associated with motor symptoms such as bradykinesia, rigidity, tremor and postural instability. Moreover, PD is generally associated with non-motor symptoms (NMSs) like olfactory and gustatory dysfunctions, sleep problems, autonomic dysregulation and neuropsychiatric symptoms, such as apathy, anxiety and cognitive impairment [1,2,3,4,5,6]. Weight change, one of the NMSs frequently neglected, usually precedes the appearance of motor symptoms in PD [7]. This change may range from weight loss to weight gain in relation to the different stages of the disease. Subjects were affected by weight loss prior to PD diagnosis, while in the first 10 years after the diagnosis they showed weight gain, and then renewed weight loss when the disease advances [8,9]. The mechanism underlying this weight change in PD patients is controversial. In particular, the weight loss can be associated to malnutrition, bone fractures, cognitive decline and worsening quality of life [10,11]. Few data are available regarding the contribution of NMSs and motor symptoms on weight change. In healthy controls the olfactory deficit, that often occurs in older age, is associated with weight loss and malnutrition [12] and a significant correlation between body mass index (BMI) and olfactory function has been observed [13,14]. Subjects with olfactory dysfunction typically show problems with food intake, reduced enjoyment in social life and become more prone to apathy and depression [15].

The aim of this study was to evaluate the role of olfactory and gustatory dysfunction, apathy, fatigue, depression and motor symptoms on body weight in PD patients.

## 2. Materials and Methods

### 2.1. Patients

We evaluated 112 participants (57 men and 55 women) in this study, 63 PD patients (mean age ± SD, 69.2 ± 10.1) and 49 healthy controls (mean age ± SD, 68 ± 9.6). In this study, 12 patients from our previous studies [16,17] were enrolled. Data collection started from September 2018 to October 2019 and further participants were recruited at the Movement Disorders Center of the University of Cagliari during regular out-patient follow-up examination. PD was diagnosed according to Gelb criteria [18] and United Kingdom Parkinson’s Disease Society Brain Bank criteria [19]. 

Controls were identified among relatives of non-Parkinsonian patients attending the out-patient department during the same period without evidence of any neurological disease. Exclusion criteria were atypical Parkinsonism, dementia, psychiatric conditions interfering with study participation, and chronic/acute rhinosinusitis. In order to evaluate weight differences between patients and controls all participants were divided into two age groups: 45–65 years (*n* = 44), and ≥ 66 years (*n* = 68).

### 2.2. Procedures 

In both PD patients and controls we collected age (years), weight (kg), height (m), BMI (kg/m^2^), olfactory and gustatory function, cognitive performance, apathy and fatigue [20]. The cognitive performance was evaluated by the Montreal Cognitive Assessment (MoCA) [6,21,22], fatigue and apathy were assessed by the Parkinson’s Disease Fatigue Scale (PFS) [23] and the Starkstein Apathy Scale (SAS) [24], respectively. Among PD patients, motor severity was evaluated by the Hoehn and Yahr (H&Y) modified scale [25], motor disability by Unified Parkinson’s Disease Rating Scale part III (UPDRS-III) [26] and therapy was assessed using levodopa equivalent daily dosage (LEDD) [27]. 

### 2.3. Olfactory Function

Olfactory function was evaluated using the Sniffin Sticks test (Burghart Messtechnik, Wedel, Germany) that considers three olfactory tasks, odor threshold (OT), odor discrimination (OD), and odor identification (OI) [16,28,29,30,31]. Participants were instructed to drink only water 1 h before the experiment, and to avoid smoking and scented products on the testing day. Sniffin Sticks are pen-like odor-dispensing devices and the complete procedure lasted 30–40 min [16,29]. Each pen (length of 14 cm and an inner diameter of 1.3 cm) was positioned at approximately 2 cm in front of both participants’ nostrils for a few seconds. All subjects were blindfolded during the OT and OD task. 

First, OT was determined using n-butanol with 16 stepwise dilutions [30]. OT was evaluated using a three-alternative forced-choice task (3AFC) and the single-staircase technique [16,28,29,31]. Scores of OT ranged from 16 (participants who were able to detect the lowest concentration of n-butanol) to 1 (participants who were unable to detect the highest concentration). 

Second, OD was assessed over 16 trials. In the OD task, three different pens were presented, two containing the same odor and the third containing the target odorant using 3AFC task. The OD score is considered as the sum of the correct responses and ranged from 0 to 16 points [32]. Third, OI was assessed using 16 common odors presented with four verbal descriptors in a multiple forced choice format (three distractors and one target).

The total score (threshold–discrimination–identification: TDI) was calculated: a value as > 30.5, ≤ 30.5 and ≤ 16.5 is considered normosmia, hyposmia and functional anosmia, respectively [31,32].

### 2.4. Gustatory Function

The gustatory function was evaluated using the Taste strips test (Burghart Messtechnik, Wedel, Germany) with four concentrations for each modality: sweet, bitter, sour and salty [33]. Before the test, the mouths of the participants were rinsed with water. The score ranged from 0 to 16 and a score < 9 was considered hypogeusia.

### 2.5. Statistical Analysis

Normal distribution of the data was assessed using the Shapiro-Wilk test. Statistical analysis was performed by the SPSS software version 22 for Windows (IBM, Armonk, N.Y., USA). Data were presented as mean values ± standard deviation. Several one-way analyses of variance (ANOVA) between subjects were carried out to evaluate statistical differences in age, weight and height. Moreover, several ANOVAs between subjects were carried out to assess differences of olfactory and gustatory function, apathy, fatigue and depression in PD patients compared to controls. 

Moreover, a multivariate linear regression analysis was performed to assess the contribution of olfactory, gustatory dysfunction, apathy, fatigue, depression, and motor symptoms on weight regulation. In the multivariate linear regression analysis, weight was a dependent variable, while olfactory, gustatory dysfunction, apathy, fatigue, depression, and motor symptoms (UPDRS and disease duration) were independent variables. The significance level was set at 0.05.

## 3. Results 

No statistical differences (*p* > 0.05) between patients and controls were observed for age and height (Table 1). Instead, for weight statistical differences between patients and controls were observed only in the age range 45–65 years (F_(1,42)_ = 4.193, *p* = 0.047).

In PD patients, the mean values for disease duration, H&Y, UPDRS and LEDD were: 4.6 ± 3.7, 2.9 ± 4.4, 23.9 ± 13.1 and 320.5 ± 284.9, respectively. Mean values of odor threshold, discrimination, identification, cognitive ability, apathy fatigue and depression in patients and controls are reported in Table 2. Significant differences between PD and controls were observed in olfactory function, apathy, fatigue and depression (Table 2). The analyses of each individual variable showed significant differences for OT (F_(1,110)_ = 20.417, *p* ≤ 0.005, partial η^2^ = 0.247), OD (F_(1,110)_ = 39.309, *p* ≤ 0.005, partial η^2^ = 0.263), OI (F_(1,110)_ = 56.155, *p* ≤ 0.005, partial η^2^ = 0.338), TDI score (F_(1,110)_ = 61.146, *p* ≤ 0.005, partial η^2^ = 0.357), apathy (F_(1,110)_ = 20.331, *p* ≤ 0.005, partial η^2^ = 0.156), fatigue (F_(1,109)_ = 24.442, *p* ≤ 0.005, partial η^2^ = 0.183) and depression (F_(1,78)_ = 33.796, *p* ≤ 0.005, partial η^2^ = 0.302). 

Our results indicated that 61.9% (*n* = 39) of PD patients had hyposmia, while 38.1% (*n* = 24) had anosmia. Weight mean values were 77.6 ± 17.7 and 71.8 ± 18.4 in these two groups, respectively. Instead, in the control group, 57.1% (*n* = 28) showed hyposmia and 42.9 % (*n* = 21) had normosmia. Weight mean values were 69.1 ± 14.5 and 67.2 ± 10.6 in subjects with hyposmia and normosmia, respectively.

As regards gustatory function, we observed a statistically significant difference between PD patients and controls (Table 3). In particular, analyses of each individual dependent variable showed significant differences between PD patients and controls for sweet (F_(1,110)_ = 18.470, *p* ≤ 0.005, partial η^2^ = 0.118), salty (F_(1,110)_ = 21.523, *p* ≤ 0.005, partial η^2^ = 0.164), sour (F_(1,110)_ = 6.499, *p* = 0.012, partial η^2^ = 0.056), bitter (F_(1,110)_ = 7.204, *p* = 0.008, partial η^2^ = 0.061) and total score (F_(1,110)_ = 24.341, *p* ≤ 0.005, partial η^2^ = 0.181) (Table 3).

Consequently, PD patients exhibited impaired olfactory and gustatory function, and were affected by higher scores of apathy, fatigue and depression compared to controls. 

In addition, multivariate linear regression analyses were performed using weight as a dependent variable in three different models. In Model 1 (Table 4) depression, UPDRS, apathy, fatigue and LEDD were independent variables. A significant contribution emerged for UPDRS, apathy and fatigue (F_(5,15)_ = 3.492, *p* < 0.05), while no contributions were found for depression and LEDD. 

A positive correlation was observed between weight and apathy (Figure 1A), while negative correlations were observed between weight, UPDRS and fatigue (Figure 1B,C). This Model explains 38% of variance (Adjusted R^2^ = 0.384). In Model 2 (Table 4) OT, OD, and OI were independent variables. A significant negative correlation was observed only for OT (F_(3,59)_ = 1.789, *p* < 0.05, Adjusted R^2^ = 0.037) (Figure 2A). Finally, in Model 3 (Table 4) sweet, salty, sour and acid perception were independent variables, while weight was a dependent variable. A significant negative correlation emerged only for sweet perception (F_(4,58)_ = 8.047, Adjusted R^2^ = 0.313, *p* < 0.01) (Figure 2B).

## 4. Discussion

Generally, as reported in previous studies [4,5,6,16,17,34] PD patients showed increased levels of apathy, fatigue and depression and exhibited impaired olfactory and gustatory function, compared to controls. Furthermore, despite weight gain or weight loss observed in relation to the stage of the disease [8,9], the mechanism of this change in PD remains still controversial. Our study, for the first time, evaluates the effect that motor symptoms and NMSs have on weight change in PD patients. In detail, our data showed a negative correlation between motor symptoms (UPDRS) and weight in PD patients. This result, in line with previous studies [7,35,36], suggests that weight loss is largely the consequence of disease progression rather than involuntary movements or a decrease in food intake. In fact, Markus and Colleagues [37] found that the muscle rigidity in PD patients was associated with higher resting-energy expenditure. However, in advanced and complicated forms of PD, the frequent presence of dyskinesia is certainly associated with greater energy expenditure. The weight change in PD patients is associated to worsening quality of life, infections, malnutrition, cognitive decline and bone fracture. Moreover, Ma and Colleagues [10] suggested that the body weight loss is related to a low density of nigrostriatal dopamine.

The mechanism of weight change in PD is considered multifactorial in relation to the various stages of the disease; moreover, physiological parameters such as neuroendocrine influences (such as orexin, ghrelin, and leptin), motivation and rewarding, hunger and satiation [38], apathy, fatigue and gustatory/olfactory deficits may influence weight and/or eating behavior. In particular, a contribution of olfactory impairment, gustatory dysfunction, and apathy can affect the food intake in the early stage of PD [9,39]. As a further confirmation, our multivariate linear regression analyses, using gustatory and olfactory function as independent variables, showed that only an impairment in OT and sweet perception play a significant role on body weight in PD patients. A possible explanation for the correlation between OT and body weight could be due to the different pathway of activation for OT, OI and OD, in view of the fact that OT could be due to individual differences of the nasal cavity [30], while OI and OD are usually associated to cognitive central pathways connecting orbitofrontal cortex, piriform cortex and amygdale. Similarly, in healthy controls a significant positive correlation between BMI and OT, but not for OI and OD, was found [13]. 

Instead, the correlation between body weight versus sweet taste could be due to compulsive eating in PD patients particularly during dopamine replacement therapy [40], in view of the fact that PD patients with compulsive eating preferred to consume sweet snacks. Similarly, an increased craving for sweet food or carbohydrates has been observed in patients with dementia and Alzheimer’s disease [41].

Likewise, in our data, a significant correlation between weight and apathy or fatigue was observed. In particular, a high score of apathy was related to high body weight, while a high score in fatigue was associated to low weight. Apathy and fatigue are two common NMSs in PD, associated to an impairment of basal ganglia and alteration of frontal-subcortical connections [42]. Apathy is an emotional deficit that can be associated to olfactory dysfunction in PD [16,43,44]. In fact, a decrease in olfactory function that occurs for a long period of time can decrease emotional memory of external stimuli, as reported in our previous work [16]. Instead, fatigue is one of the NMSs that affects about 50% of PD patients [10,20] and seems to be related to anxiety and apathy [20,45]. 

Furthermore, statistical differences in weight between PD patients and controls were found only in the 45–65 years age range, while no statistical differences were observed in elderly patients. These results are in line with previous studies [35,46], which reported no statistical differences in weight between elderly PD patients and controls. In particular, Lorefält and Colleagues [35] suggested that elderly PD patients usually exhibited a weight loss associated to low physical activity. Instead, in healthy subjects, other previous studies [47,48] indicated that the body mass index increased in relation to the age.

In our data no correlation was found between LEDD and weight. Instead, previous studies [9,40] suggest that therapy using dopamine agonist could induce compulsive eating in PD patients. A possible explanation could be due to different stages of disease under evaluation. 

Recently, previous studies [49,50] evaluated the role of genetic factors such as odorant-binding protein IIa (OBPIIa) and the 6-n-propylthiouracil (PROP) taster status mediated by the TAS2R38 locus in PD patients. In particular, PROP status is associated with numerous diseases not only correlated to taste function [49], while OBPIIa plays an important effect in the odors perception [50].”

In view of the fact that the body weight regulation in PD is a complex and multifactorial process, which involves many aspects such as gastrointestinal dysfunction, dysphagia, genetic factors, constipation and gastroparesis associated to nausea and vomit, further studies are necessary to evaluate the role of these other NMSs in weight change. 

## 5. Conclusions

In detail, our results reported a significant contribution of apathy, fatigue, OT, sweet perception, and UPDRS on body weight in PD patients. This study may provide better knowledge in PD patients on the mechanism underlying food intake and weight regulation. An early identification of patients with weight problems may help to develop new strategies to prevent malnutrition.

## Figures and Tables

**Figure 1 brainsci-10-00218-f001:**
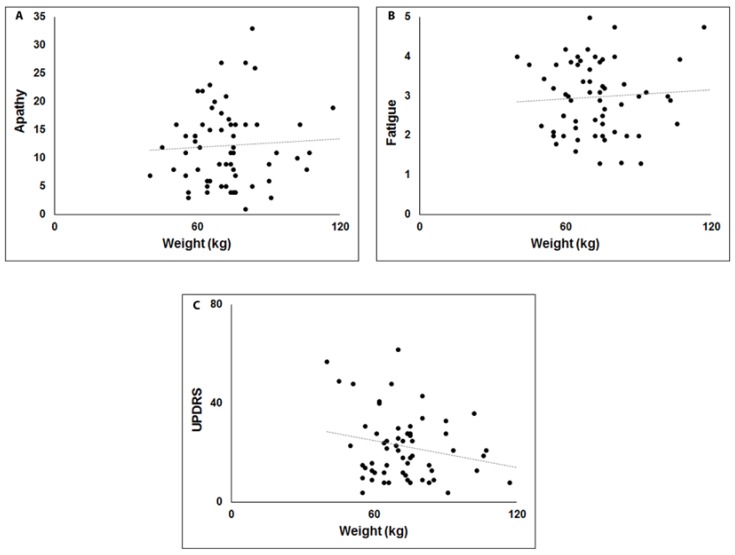
Scatterplots of the relationship between body weight (kg), apathy (**A**), fatigue (**B**) and versus motor disability (UPDRS) (**C**).

**Figure 2 brainsci-10-00218-f002:**
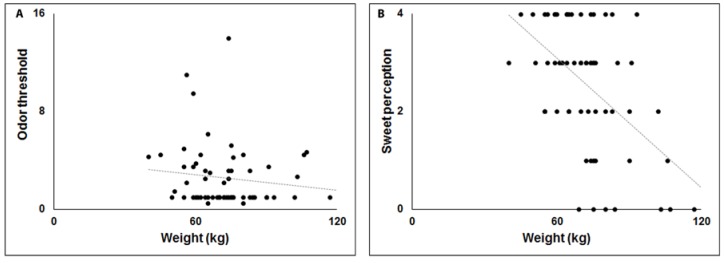
Scatterplots of the relationship between body weight (kg), odor threshold (**A**) and sweet perception (**B**).

**Table 1 brainsci-10-00218-t001:** Demographic and clinical information of all participants.

	Controls (Mean ± SD)	PD (Mean ± SD)	Significance
Age (years)	67.9 ± 9.6	69.2 ± 10.1	*p* = 0.503
Height (m)	1.63 ± 0.104	1.64 ± 0.103	*p* = 0.442
Weight (kg) 45–65 years	68.2 ± 12.2	78.9 ± 22.03	***p* = 0.047**
Weight (kg) ≥ 66 years	67.72 ± 14.01	70.4 ± 13.133	*p* = 0.433

Legend: SD = standard deviation. Bold indicated statistical differences between patients and controls.

**Table 2 brainsci-10-00218-t002:** Statistical differences of olfactory function, cognitive ability, apathy, fatigue, and depression between Parkinsonian patients (PD) and controls.

	Controls (Mean ± SD)	PD (Mean ± SD)	Significance
Threshold	5.6 ± 4.4	2.6 ± 2.5	***p* ≤ 0.005**
Discrimination	10.8 ± 2.7	7.3 ± 3.1	***p* ≤ 0.005**
Identification	12.1 ± 2.6	7.6 ± 3.5	***p* ≤ 0.005**
TDI score	28.6 ± 7.3	17.6 ± 7.3	***p* ≤ 0.005**
Cognitive ability	25.9 ± 3.7	21.4 ± 5.8	***p* ≤ 0.005**
Apathy	7.3 ± 3.7	12.5 ± 6.9	***p* ≤ 0.005**
Fatigue	2.1 ± 0.70	2.9 ± 0.9	***p* ≤ 0.005**
Depression	5.4 ± 5.1	14.3 ± 8.7	***p* ≤ 0.005**

Legend: SD = standard deviation. Bold indicated significant level *p* < 0.05.

**Table 3 brainsci-10-00218-t003:** Statistical differences of gustatory function between parkinsonian patients (PD) and controls.

	Controls (Mean ±SD)	PD (Mean ± SD)	Significance
Sweet	3.3 ± 0.8	2.5 ± 1.3	***p* ≤ 0.005**
Salty	3.1 ± 1.1	1.9 ± 1.5	***p* ≤ 0.005**
Sour	2.3 ± 1.5	1.7 ± 1.2	***p* = 0.012**
Bitter	2.6 ± 1.5	1.9 ± 1.5	***p* = 0.008**
Total taste score	11.3 ± 3.1	8 ± 3.9	***P* ≤ 0.005**

Legend: SD = standard deviation. Bold indicated significant level *p* < 0.05.

**Table 4 brainsci-10-00218-t004:** Multivariate linear regression analyses in patients with Parkinson’s disease using weight as a dependent variable.

	Unstandardized Coefficients		Standard Coefficients		
	B	Std Error	β	t	Significance
**Model 1**					
Fatigue	−12.562	5.314	−0.726	−2.364	**0.032**
Apathy	0.815	0.358	0.501	2.275	**0.038**
UPDRS	−0.386	0.171	−0.492	−2.260	**0.039**
Depression	0.454	0.397	0.290	1.143	0.271
LEDD	0.010	0.013	0.168	0.752	0.463
**Model 2**					
Threshold	−0.781	0.346	−0.287	−2.257	**0.028**
Discrimination	0.302	0.863	0.055	0.350	0.728
Identification	0.384	0.758	0.080	0.506	0.615
**Model 3**					
Sweet	−5.375	1.625	−0.426	−3.308	**0.002**
Salty	−0.249	1.356	−0.022	−0.184	0.855
Sour	−0.851	1.558	−0.060	−0.546	0.587
Bitter	−2.444	1.360	−0.222	−1.797	0.078

Legend: Levodopa equivalent daily dosage (LEDD); Unified Parkinson’s Disease Rating Scale (UPDRS). Bold indicated significant level *p* < 0.05.

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
