# Peer review of "Effect of Olfactory and Gustatory Dysfunction and Motor Symptoms on Body Weight in Patients with Parkinson’s Disease"

_brainsci, 2020, doi:10.3390/brainsci10040218_

Round 1
Reviewer 1 Report
The submitted manuscript presented solid data supporting that olfactory and gustatory dysfunction, apathy, fatigue, depression and motor symptoms have significant impact on the body weight of patients. The study is overall well described and conducted. Although a longitudinal study could significant help to further understand how these factors could affect the weight during the disease progress, the data set collected is overall solid enough to support the conclusions made by the authors.
One of the major weakness of this manuscript is that the author showed significant difference caused by the factors listed above, and that these factors all positively correlate with PD, yet no significant difference in weight is observed between PD and healthy controls. It is described in the manuscript that previous literature showed similar results, and it is understandable that the patient body weight is affected by multiple factors that are complicate to study. However, the lack of further discussion about why the body weight is not significantly changed in PD patients, while all the factors listed above are significantly altered in PD patients, presents a major logical weakness for this paper. For example, in the manuscript, it mentioned that 50% of patients showed NMS such as fatigue, which is shown by the authors to be negatively correlates with body weight. One would assume that this the 50% PD patients with fatigue would show significantly decreased body weight compared to the control, however, from the manuscript, it is hard for the readers to tell whether that’s the case or not. This weakness can be easily addressed by better stratifying the PD patients and compare them to the healthy controls.
Minor points:
- The significance labeled in each table should be consistent. The P values should be provided as actual number in all data sets.
- PD is known to be associated with various of genetic factors that could potentially affect metabolism. While it is out of the scope of this study, it is worth to discuss the possible influence of genetic background in weight.
- A brief discussion about how weight change could in turn affect the PD progression and patient performance would strengthen the manuscript and warrant future longitudinal studies.
- In line 117, the author wrote “Our results indicated that 61.9% (n=39) of PD patients had hyposmia, while 38.1% (n=24) had anosmia”. Is there a reason why such numbers are not listed for the control group?
- In line 176, the authors stated that fatigue is related to apathy. If this is true, then it would be inappropriate to make apathy and anxiety as independent factors, and the authors should better justify their analysis in table 4.
Author Response
Dear Editor of Brain Sciences
Herewith, we submit the revised manuscript entitled “Effect of olfactory and gustatory dysfunction and motor symptoms on body weight in patients with Parkinson’s disease”.
We have complied with the Referees’ requests and our revisions are indicated in the manuscript in red. We hope that our work is now suitable for publication in Brain Sciences.
Many thanks for your consideration.
Your Sincerely,
Carla Masala
Reviewers comments and Author’s answers:
Reviewer 1
a) The submitted manuscript presented solid data supporting that olfactory and gustatory dysfunction, apathy, fatigue, depression and motor symptoms have significant impact on the body weight of patients. The study is overall well described and conducted. Although a longitudinal study could significant help to further understand how these factors could affect the weight during the disease progress, the data set collected is overall solid enough to support the conclusions made by the authors.
One of the major weakness of this manuscript is that the author showed significant difference caused by the factors listed above, and that these factors all positively correlate with PD, yet no significant difference in weight is observed between PD and healthy controls. It is described in the manuscript that previous literature showed similar results, and it is understandable that the patient body weight is affected by multiple factors that are complicate to study. However, the lack of further discussion about why the body weight is not significantly changed in PD patients, while all the factors listed above are significantly altered in PD patients, presents a major logical weakness for this paper. For example, in the manuscript, it mentioned that 50% of patients showed NMS such as fatigue, which is shown by the authors to be negatively correlates with body weight. One would assume that this the 50% PD patients with fatigue would show significantly decreased body weight compared to the control, however, from the manuscript, it is hard for the readers to tell whether that’s the case or not. This weakness can be easily addressed by better stratifying the PD patients and compare them to the healthy controls.
Answer a) We thank the Reviewer for this observation. According to the Reviewer’s observation, we performed one-way between subjects ANOVA to evaluate weight differences between patients with Parkinson’s disease and healthy controls.
In the Patients section we included this sentence: “In order to evaluate weight differences between patients and controls all participants were divided into two age groups: 45–65 years (N=44), and ≥ 66 years (N=68).”
Moreover, the Results section was changed in the following: “No statistical differences (p>0.05) between patients and controls were observed for age and height (Table 1). Instead, for weight statistical differences between patients and controls were observed only in the 45–65 years age range (F(1,42)=4.193, p=0.047). Consequently, the Table 1 was changed in the Manuscript.”
Finally, in the Discussion section we reported: “Furthermore, statistical differences in weight between PD patients and controls were found only in the 45-65 years age range, while no statistical differences were observed in elderly patients. These results are in line with previous studies [46,47], which reported no statistical differences in weight between elderly PD patients and controls. In particular, Lorefält and Colleagues [46] suggested that elderly PD patients usually exhibited a weight loss associated to low physical activity. Instead, in healthy subjects other previous studies [48,49] indicated that the body mass index increased in relation to the age.”
Minor points:
b) The significance labeled in each table should be consistent. The P values should be provided as actual number in all data sets.
Answer b) According the Reviewer suggestion in the Results section the following sentences were changed in red on the text: “The analyses of each individual variable showed significant differences for OT (F(1,110)=20.417, p≤0.005, partial η2=0.247), OD (F(1,110)=39.309, p≤0.005, partial η2=0.263), OI (F(1,110)=56.155, p≤0.005, partial η2=0.338), TDI score (F(1,110)=61.146, p≤0.005, partial η2=0.357), apathy (F(1,110)=20.331, p≤0.005, partial η2=0.156), fatigue (F(1,109)=24.442, p≤0.005, partial η2=0.183) and depression (F(1,78)=33.796, p≤0.005, partial η2=0.302).” and “The analyses of each individual variable showed significant differences for OT (F(1,110)=20.417, p≤0.005, partial η2=0.247), OD (F(1,110)=39.309, p≤0.005, partial η2=0.263), OI (F(1,110)=56.155, p≤0.005, partial η2=0.338), TDI score (F(1,110)=61.146, p≤0.005, partial η2=0.357), apathy (F(1,110)=20.331, p≤0.005, partial η2=0.156), fatigue (F(1,109)=24.442, p≤0.005, partial η2=0.183) and depression (F(1,78)=33.796, p≤0.005, partial η2=0.302).”
Moreover, the Tables 1 and 2 were revised according the Reviewer’s suggestion in red on the Manuscript.
c) PD is known to be associated with various of genetic factors that could potentially affect metabolism. While it is out of the scope of this study, it is worth to discuss the possible influence of genetic background in weight.
Answer c) We thank the Reviewer for this interesting observation, the following sentences were included in the Discussion section: “Recently previous studies (Cossu et al., 2018; Melis et al., 2019) evaluated the role of genetic factors such as odorant-binding protein IIa (OBPIIa) and the 6-n-propylthiouracil (PROP) taster status mediated by the TAS2R38 locus in PD patients. In particular, PROP status is associated with numerous diseases not only correlated to taste function (Cossu et al., 2018), while OBPIIa plays an important effect in the odors perception (Melis et al., 2019).”
d) A brief discussion about how weight change could in turn affect the PD progression and patient performance would strengthen the manuscript and warrant future longitudinal studies.
Answer d) In line with the Reviewer’s suggestion, the following discussion was included in the text: “The weight change in PD patients is associated to worsening quality of life, infections, malnutrition, cognitive decline and bone fracture. Moreover, Ma and Colleagues [10] suggested that body weight loss is related to a low density of nigrostriatal dopamine.”
e) In line 117, the author wrote “Our results indicated that 61.9% (n=39) of PD patients had hyposmia, while 38.1% (n=24) had anosmia”. Is there a reason why such numbers are not listed for the control group?
Answer e) We thank the Reviewer for the observation, Authors included the following description in the Results section of the Manuscript: “Instead, in the control group, 57.1% (n=28) showed hyposmia and 42.9 % (n=21) had normosmia. Weight mean values were 69.1 ± 14.5 and 67.2 ± 10.6 in subjects with hyposmia and normosmia, respectively.”
f) In line 176, the authors stated that fatigue is related to apathy. If this is true, then it would be inappropriate to make apathy and anxiety as independent factors, and the authors should better justify their analysis in table 4.
Answer f) The multivariate linear regression analysis was performed to assess the contribution of olfactory, gustatory dysfunction, apathy, fatigue, depression and motor symptoms on weight regulation. In the multivariate linear regression analysis, weight was a dependent variable, while olfactory, gustatory dysfunction, apathy, fatigue, depression and motor symptoms (UPDRS and disease duration) were independent variables.
The application of the multivariate linear regression appears appropriate to assess the effect of different parameters on weight regulation in PD patients.
Editorial Office of Brain Sciences
Because the reviewers have suggested that your manuscript should undergo extensive English editing, please address this during revision. We suggest that you have your manuscript checked by a native English speaking colleague or use a professional English editing service.
Answer) The Manuscript was revised by Dr Alan Moat, an Author, who is a native English speaking Colleague from England. All corrections were in red color in the text.

Reviewer 2 Report
Author suggest olfactory dysfunction could contribute body weight which is another type of non motor symptoms in PD
However, there is even doubt about demographic data including stage, duration, subtype of PD, non motor scale. Weight and Height omit its measurement unit. For example, H&Y stage suggested as 2.9 ± 4.4
So, there can be possible that terminal stage PD patient included, but UPDRS score was only 23.9 ± 13.1. It could be possible? this means that maximum UPDRS score was within 43.
Also, there was no regression analysis in control group. Author's suggestion cannot be specific in PD group. Must analyze in both group.
Author Response
Dear Editor of Brain Sciences
Herewith, we submit the revised manuscript entitled “Effect of olfactory and gustatory dysfunction and motor symptoms on body weight in patients with Parkinson’s disease”.
We have complied with the Referees’ requests and our revisions are indicated in the manuscript in red. We hope that our work is now suitable for publication in Brain Sciences.
Many thanks for your consideration.
Your Sincerely,
Carla Masala
Reviewers comments and Author’s answers:
Reviewer 2
Author suggest olfactory dysfunction could contribute body weight which is another type of non motor symptoms in PD.
g) However, there is even doubt about demographic data including stage, duration, subtype of PD, non motor scale. Weight and Height omit its measurement unit. For example, H&Y stage suggested as 2.9 ± 4.4
Answer g) According to the Reviewer observation, the measurement unit was included in the Material method section and in Table 1.
As regards the H&Y stage, Authors indicated a mean score ± standard deviation. The H&Y scale describes five stages to PD progression according the description reported in literature (Hoehn, M.M.; Yahr, M.D. Parkinsonism: onset, progression, and mortality. Neurology 1967, 50(2), 318–318. https ://doi.org/10.1212/WNL.50.2.318).
h) So, there can be possible that terminal stage PD patient included, but UPDRS score was only 23.9 ± 13.1. It could be possible? this means that maximum UPDRS score was within 43.
Answer h) The mean values of UPDRS- III were in line to previous data indicated in literature (Masala et al., 2018; Solla et al., 2019; Fahn, S.; Elton, R.; and Members of the UPDRS Development Committee (1987) The Unified Parkinson’s Disease Rating Scale.)
In particular, in our previous papers were reported the following mean values ± standard deviation for Unified PD Rating Scale (UPDRS) 18.1 ± 9.6 (Solla et al., 2019) and 17.9 ± 9.5 (Masala et al., 2018). The UPDRS score ranges from 0 to 108.
i) Also, there was no regression analysis in control group. Author's suggestion cannot be specific in PD group. Must analyze in both group.
Answer i) Many other previous papers evaluated the role of olfactory/gustatory dysfunction, apathy, fatigue and depression on weight change in healthy controls (Fluitman et al., 2019; Skrandies et al., 2015; Patel et al., 2015). Consequently, this was not the aim of our Manuscript. The aim of our study was to evaluate the role of olfactory and gustatory dysfunction, apathy, fatigue, depression and motor symptoms on body weight in PD patients.
Editorial Office of Brain Sciences
Because the reviewers have suggested that your manuscript should undergo extensive English editing, please address this during revision. We suggest that you have your manuscript checked by a native English speaking colleague or use a professional English editing service.
Answer) The Manuscript was revised by Dr Alan Moat, an Author, who is a native English speaking Colleague from England. All corrections were in red color in the text.
